# Study on the Stabilization of a New Type of Waste Solidifying Agent for Soft Soil

**DOI:** 10.3390/ma12050826

**Published:** 2019-03-11

**Authors:** Jiansheng Shen, Yidong Xu, Jian Chen, Yao Wang

**Affiliations:** 1School of Civil Engineering and Architecture, Ningbo Institute of Technology, Zhejiang University, Ningbo 315100, China; sjs@nit.zju.edu.cn; 2School of Civil Engineering and Architecture, Jiangsu University of Science and Technology, Zhenjiang 212000, China; cj199443@163.com (J.C.); wy903961287@163.com (Y.W.)

**Keywords:** DGSC-solidified soil, soil stabilization, unconfined compressive strength, quasi-water-cement ratio

## Abstract

The use of desulfurization gypsum and steel/furnace slag composite cementitious material (DGSC) to solidify soft soil can fully utilize industrial wastes, reduce cement use and protect natural resources. By studying the unconfined compressive strengths of DGSC-solidified soil with different mix ratios, water-binder ratios and curing periods, the influence of those factors on the unconfined compressive strength of the soil can be analyzed. Furthermore, the quasi-water-cement ratio is introduced to predict the strength of the DGSC-solidified soil. The results show that the higher the DGSC content is, the better its effect on the soft soil. The variation in the unconfined compressive strength of DGSC-solidified soil overtime can be described by the same trend as that of cement-solidified soil but its early strength is lower than that of cement-solidified soil. When the water-binder ratio of the DGSC-solidified soil is the same as that of the cement-solidified soil, after a28-day curing period, the content of DGSC is higher than that of the 5% cement content, so the DGSC solidification effect is comparable to that of cement. Therefore, using DGSC instead of cement as a soft soil solidifying agent can meet the strength requirements of solidified soil.

## 1. Introduction

Soft soils encountered in construction work pose a considerable threat to the long-term performance and operation of facilities the foundation or underlying structure of which is made of these materials [1,2,3,4,5,6]. Due to the growth in the population and the lack of stable ground for the construction of civil projects, it is vital to improve soft and problematic soil deposits through appropriate stabilization methods. Soil reengineering involves mechanical, chemical and biochemical procedures. Through the application of these methods, soft soils improve in strength, density and durability [7,8,9,10,11,12,13]. Thus far, Portland cement and other chemical additives and chlorides have been the only materials utilized as binders or pozzolans to improve the mechanical properties of soft or expansive soils. Due to the use of these materials, there has been an increase in CO_2_ emissions into the atmosphere, consequently contributing to global warming; for each ton of Portland cement used in construction work, an equivalent amount of CO_2_ is released into the atmosphere. Research to find an alternative to Portland cement has been ongoing [14].

Experts have applied various materials to improve the strength, density and durability of soil while lowering CO_2_ emissions. Such materials include oyster shell powder, quarry dust, snail shell dust, sawdust, crushed ceramics, crushed glasses, crushed plastics and fly ash respectively [15,16,17,18,19,20,21,22,23]. Other possible alternatives are the use of alkali-activated binders using industrial by-products comprising silicate materials [24,25]. Ground granulated blast furnace slag is one of the most common industrial by-products used as binder materials due to its outstanding performance [26,27,28,29]. Steel slag is the main waste of the steel industry. Due to the similarity in the chemical compositions (C_3_S, C_2_S, C_2_F, Fe_2_O_3_ and RO phases; solutions of solid CaO, MgO, MnO and FeO) of steel slag and ordinary Portland cement, steel slag has great application prospects for cement and concrete [30,31,32]. However, during the hydration reaction, the potential pozzolanic activity of the slag is activated in the alkaline environment, the hydration reaction of cement accelerates and the ettringite produced from the reaction of the gypsum and the hydrated calcium aluminate contributes to the improvement of the strength and sample density [33,34,35]. Therefore, they can replace Portland cement as an alternative cementing material or supplementary cementing material. Such materials are all derivatives of solid wastes and sustainably serve the purpose of soft soil reengineering. Furthermore, the synthesis of geopolymer cements with the derivatives of these solid waste materials has been tested for use as base material [12].

In this paper, steel slag, desulfurized gypsum and furnace slag are used as the main raw materials and cement clinker is added as a mineral activator to make desulfurization gypsum, steel slag and ground granulated blast furnace slag compound cementitious material (hereinafter referred to as DGSC). This new type of material is used to improve the material properties of soft soils. Through a comparison of the unconfined compressive strength of DGSC-solidified soil and the unconfined compressive strength of cement-stabilized soil, the mechanical properties of the solidified soft soil are discussed and the curing effect of the DGSC curing agent is predicted with a water-cement ratio index to explore the feasibility of DGSC in practical engineering applications.

## 2. Experiments

### 2.1. Raw Materials

The curing agent (DGSC) is mixed with the excitation agent (A), desulfurization gypsum (DG), steel slag (SS) and ground granulated blast furnace slag (S). The composition ratio is m(DG):m(SS):m(S):m(A) = 12:43:40:5. Conch 42.5R ordinary Portland cement is used; desulfurized gypsum is obtained from the industrial waste of the Ningbo Beilun Power Plant (Ningbo, China). The specific surface area of the ground granulated blast furnace slag, produced by Baotian New Building Material Co., Ltd. (Shanghai, China), is 400 m^2^/kg. The specific surface area of the steel slag, which is obtained by grinding the solid waste discharged from Shanghai BaoSteel Co., Ltd. (Shanghai, China), is 450 m^2^/kg. The excitation agent is composed of fine cement clinker powder and sulphoaluminate micro powder. The main chemical composition of each raw material in the DGSC is shown in Table 1. The physical and mechanical properties of the curing agent are shown in Table 2. The experimental soil is the typical silty clay in Ningbo and its physico-mechanical indexes are shown in Table 3.

To study the role of the components of the DGSC in terms of its strength development, we designed the mix proportion for the following orthogonal experiment (see Table 4) and obtained the physico-mechanical parameters of the curing agent with different mix proportions (see Table 5).

Through orthogonal analysis, it was found that the content of ground granulated blast furnace slag contributed the most to the DGSC strength, followed by the activator composition; the desulfurization gypsum had a minimal contribution to the DGSC strength.

### 2.2. Preparation Process

In this paper, the effect of DGSC curing is studied by comparing the unconfined compressive strengths of cement-solidified soil and DGSC-solidified soil samples at different curing periods under different curing agent incorporation ratios and water-cement ratios. The incorporation ratios of the samples are 8%, 10%, 12%, 15% and 20%. The water-binder ratio of the cement-solidified soil is 0.3 and the water-binder ratios of the DGSC-solidified soilare 0.3, 0.4 and 0.5. Here, the incorporation ratio (aw) refers to the content of the curing agent in the solidified soil, which is the ratio of the mass of the incorporated solid material to the mass of the wet soil. The amounts of each material are shown in Table 6.

#### 2.2.1. Unconfined Pressure Test Specimen Preparation

When preparing the sample, the previously prepared soil sample and curing agent are first put into the vessel and mixed manually with a spatula for 2–3 min. The weighed water is then poured into the slurry mixing pot with the mixture and stirred. After stirring, the samples are made. The unconfined compressive strength test mold (ϕ 39.1 mm, H 80 mm, three-hole mold) is made, washed clean and dried and a thin layer of Vaseline is then applied to the outer wall to facilitate the removal of the mold.

The sample preparation is as follows:

(1) The three-hole mold should be installed first.

(2) The mixture is mixed three times. After each addition, the vibrating table is used to vibrate the sample (Figure 1). Finally, the top is wiped.

(3) Two days later, under room conditions, the mold is removed and the sample is placed in a standard curing box and cured fora specified period (Figure 2).

#### 2.2.2. DGSC Curing Agent Micro Specimen Preparation

The curing agent slurry test block is put into the standard maintenance box (temperature (20 ± 1) °C, relative humidity greater than 90%) for maintenance. At the specified period (3 days, 7 days or 28 days), the sample is crushed, the center piece is removed and hydration with anhydrous ethanol is terminated. When the sample is ready, it is taken out and left to rest until the alcohol evaporates. A flat sample approximately 1 cm in diameter is selected and stored in a Zip lock bag and at the same time, another sample is ground into a powder with a mortar. These samples were also sealed in a zip lock bag and submitted to X-ray diffraction (XRD) analysis and scanning electron microscopy (SEM) analysis within two hours of sample preparation.

### 2.3. Test Methods

#### 2.3.1. Unconfined Compressive Strength Test

Due to the high strength of the solidified soil, a triaxial shear tester is used for the unconfined compression test (Figure 3) and the confining pressure is selected. The specific steps under σ_3_ = 0 are as follows:

After the sample is cured to the specified age, the sample is removed and the diameters of the upper and lower ends and the middle are measured with a Vernier caliper (Shenhan measuring tools Co., Ltd., Ningbo, China); the average of these three measurements is taken as the diameter of the sample. The heights of the sample are measured in four orthogonal directions and the average of the four heights is taken as the height of the sample. Electronic weighing (Shanghai precision instrument Co., Ltd., Nanjing, China) is used to measure the sample weight. The area and volume are then calculated.

Vaseline is applied on both ends of the sample and the sample is installed in the triaxial shear tester (Nanjing water conservancy and electric power instrument engineering Co., Ltd., Nanjing, China); the loading speed is adjusted to 0.8 mm/min. After the instrument is started, the corresponding stress-strain value is recorded by the data collector. The motor is switched off when the load no longer changes or decreases.

The destruction characteristics of the specimen are described and the damaged specimen is removed from the instrument.

According to the deformation of each stage, the stress range of each deformation stage is obtained and the breaking strength qu is obtained.

#### 2.3.2. SEM Test

A microscopic morphology test was conducted using an S-4800 field emission scanning electron microscope produced by Hitachi, Japan (Tokyo, Japan). The test surface of the sample is subjected to a gold spray treatment before the test. The SEM specifications are as follows: secondary electron resolutions 1.0 nm (15 kV) and 2.0 nm (1 kV), back scattered electron resolution 3.0 nm (15 kV), acceleration voltage 0.5–30 kV, cold field emission electron source and magnification range 30 to 800,000 times.

#### 2.3.3. Polycrystalline XRD Test

The prepared powder samples were subjected to XRD (Davinci, Bruker, Germany) and analyzed using JADE6.5 software. The X-ray diffractometer technical specifications are as follows: Cu target X-ray tube, voltage ≤50 kV, current ≤40 mA, nine automatic samplers, dual optical system.

## 3. Test Results and Analysis

### 3.1. Change Law of the Unconfined Compressive Strength of Solidified Soil with Time

Figure 4 shows the curve of the unconfined compressive strength as a function of age for solidified clays with different mix ratios and different water-cement ratios. This figure shows that the laws of the unconfined compressive strengths of DGSC-solidified soil and cement-solidified soil are consistent and increase with the curing period. The early strength of DGSC-solidified soil is lower than that of cement-solidified soil and the curing effect of DGSC before 7 days is not obvious. After the 7-day curing period, the unconfined compressive strength of DGSC-solidified soil clearly increases; after the 28-day curing period, the intensity continues to increase strongly with increasing time, while the rate of increase gradually decreases. This figure also indicates that the higher the curing agent content is, the faster the increase in the unconfined compressive strength.

Table 7 shows the effect of different water-binder ratios on the unconfined compressive strength of solidified soil for the same DGSC content. When the water-binder ratios are 0.3, 0.4 and 0.5, the 28-day unconfined compressive strengths for a 15% DGSC content are 1944 kPa, 1439 kPa and 1441 kPa, respectively. The greater the water-binder ratio is, the lower the unconfined compressive strength of the corresponding solidified soil. In the later curing period of solidified soil (after 60 days), the strength of the solidified soil with a 15% DGSC content and water-binder ratio of 0.3 is 2836 kPa, which is comparable to that with a water-cement ratio of 0.5.

When the water-binder ratio of DGSC-solidified soil is 0.3 (as shown in Figure 4a), the 28-day unconfined compressive strength for the DGSC contents of 8%, 10%, 12%, 15% and 20% are 481 kPa, 710 kPa, 1176 kPa, 1944 kPa and 3035 kPa, respectively. Increasing the binder content can improve the strength of solidified soil (Figure 4b,c). When the water-binder ratios are 0.3, 0.4 and 0.5, the 28-day unconfined compressive strength at 20% DGSC content is 3035 kPa, 2620 kPa and 2451 kPa, respectively. Additionally, the increase in the water-cement ratio is inversely proportional to the increase in strength.

When the water-binder ratio of solidified soil is 0.5 (as shown in Figure 4d), after the 14-day curing period (28-day, 60-day, and 90-day curing periods), the unconfined compressive strengths of the 20% DGSC content solidified soil are 1942 kPa, 2706 kPa, 2961 kPa and 3105 kPa, respectively. Again, the unconfined compressive strengths of the corresponding 15% cement-solidified soil are 1822 kPa, 2451 kPa, 2854 kPa and 3013 kPa, respectively. The strengths of the solidified soil during the same curing period for the first two abovementioned cases are similar. After the 28-day curing period, the unconfined compressive strength of the 15% DGSC content solidified soil is similar to that with 10% cement content. When the water-binder ratio and the curing conditions are the same, after the 28-day curing period, the content of DGSC is greater than that at 5% cement content, so the solidification effect of DGSC is comparable to that of cement. During the curing period (7–14 days), Figure 4 clearly shows that the slope of the curve of the DGSC-solidified soil is significantly higher than that of the cement-solidified soil and after the 28-day curing period, not only the strength of the solidified soil increases but also the rates of increase in the strengths of the DGSC-solidified soil and cement-solidified soil become more similar. However, before 28 days, the strength of DGSC-solidified soil is still lower than that of cement-solidified soil.

The results show that during the 7-day curing period, the hydration reaction of the cement-solidified soil was intense and the strength increased rapidly; after the 7-day curing period, the hydration reaction continued but the material content with high hydration activity was decreased slowly, so the rate of increase in the strength declined accordingly. During the 7-day curing period, because of the slow dissolution of hydration components, the unfavorable chemical composition and the soil structure clearly influenced the strength of the solidified soil and the strength of the DGSC-solidified soil increased very slowly. After the 7-day curing period, the hydration components of the desulfurization gypsum, steel slag and ground granulated blast furnace slag in the curing agent were dissolved fully and the hydration reaction was accelerated; then, the strength of the DGSC-solidified soil increased rapidly. Again, after the 28-day curing time, the material content with high hydration activity in the DGSC-solidified soil decreased slowly, so the rate of increase in strength also declined accordingly.

### 3.2. Stress-Strain Relationship of DGSC-Solidified Soil

Figure 5 shows the stress-strain relationship for solidified soils with different curing agent contents and curing periods. Figure 5a shows that the stress-strain curves for curing times of 7 days and 14 days at 8% DGSC content are parallel. As the axial strain ε_a_ increases, no significant turning point appears. This result shows that for the solidified soil with an 8% DGSC content, the early stress increase is not obvious but the deformation is large and the resulting deformation is unrecoverable plastic deformation. At the same time, for the 7-day, 60-day and 90-day curing periods, the curves are significantly elevated and an inflection point appears. At 28 days, the inflection point of the curve indicates a shift to a more linear elasticity relationship within a certain strain range.

For the solidified soil with a 10% DGSC content, as shown in Figure 5b, its stress-strain curve is similar to that of the 8% content at 7 days but slightly elevated at 14 days. This result shows that with the increase in the curing agent content, the early initial modulus E0 of the sample increases slightly. The 28-daycurve has an earlier shift than the inversion point at 8%. In the same strain range, the stress clearly increases and the initial modulus E0 increases significantly. For the curves of 60 days and 90 days, the soil behavior gradually changes to brittle failure after reaching the ultimate strength.

For the solidified soil with a DGSC content greater than or equal to 12%, Figure 5c–e shows that the staged stress-strain relationship of the DGSC-solidified soil is reached after the 14-daycuring period. The first stage is the initial stage of loading and the curve is approximately linear; the second stage of the curve is the non-linear rising stage and the stress gradually increases and reaches the peak value; the third stage of the curve is the descending stage, that is, the failure stage of the specimen. Figure 5b–e shows that the more DGSC curing agent is used, the more obvious the inflection point between the first stage and the second stage.

Through the analysis of the experimental results, the early strength of the DGSC-solidified soil is relatively low, the stress-strain curve is relatively flat, there is no obvious inflection point and the soil deformation is in the plastic state; with time, the stress-strain characteristics of the solidified soil change, the failure strain gradually decreases and the non-linear rising stage and the stress decreasing stage of the stress-strain curve are more obvious after reaching the limit. With the increase in the mix ratio, after the 28-day curing period, the soil behavior shifts from plastic failure to brittle failure.

### 3.3. DGSC Curing Mechanism Analysis

The XRD analysis results of the hydration products of the samples at different hydration times are shown in Figure 6. The hydration products are mainly ettringite and residual dihydrate gypsum. As the time of hydration is prolonged, the intensity of the diffraction peak of ettringite increases and the intensity of the diffraction peaks of dihydrate gypsum decreases. This result shows that during hydration, desulfurized gypsum is gradually consumed, the content of hardened gypsum gradually decreases and increasingly more hydrated ettringite is produced.

The SEM measurement results of the sample hydration products are shown in Figure 7 Corresponding to the results of the XRD measurements, the hydration products are mainly needle-like ettringite and pore-filling C-S-H gel in the ettringite. With the extension of the hydration time, hydration continues and an increasing amount of hydration products are produced, making the internal structure of the slurry dense.

At 3 days, Figure 7a shows that after DGSC hydration, a large amount of sheet-like and needle-like substances was produced. According to the morphological characteristics of the hydration product, this hexagonal plate-like crystal can be confirmed as Ca(OH)_2_. An analysis of the structure in the figure suggests that a large number of acicular, fibrous, coarse acicular and rod-like hydration products are produced in the paste sample block. In general, the needle-like or fibrous C-S-H gels are at tapulgite, having thick needles or rods with straight ends, no bifurcation and uniform thickness.

At 28 days, Figure 7b shows that as the time increases, the fibrous hydration products increase significantly and interdigitate and the lamellar and flaky crystal surfaces are mostly covered by these hydration products.

A comprehensive macroscopic XRD and microscopic SEM analysis shows that in the DGSC system, after the mixture is mixed with water, calcium hydroxide and other mineral phases in the steel slag are successively dissolved under the action of the activator.

The alkalinity of the system rapidly increases and the calcium-rich furnace slag begins to dissolve. In the furnace slag, active calcium oxide, alumina, silica and other components are dissolved and the sulfate in the desulfurized gypsum forms ettringite crystals.

The late increase in strength of the cementitious materials mainly depends on the formation of a large number of C-S-H gels. During hydration, C-S-H gel are produced and fill the pores of the needle-like ettringite so that the slurry is continuously densified and the remaining desulfurized gypsum is finally encapsulated by the hydration product, which functions as micro-aggregate filling. The desulfurized gypsum generates a large amount of needle-like ettringite on the surface of the hydration process, greatly improving the affinity of the desulfurized gypsum and the hydration product interface, tightly binding the desulfurized gypsum and the hydration product.

## 4. DGSC-Solidified Soil Strength Prediction

### 4.1. Relationship between the Water-Cement Ratio and the Unconfined Compressive Strength of Reinforced Soil

Currently, many scholars have studied the relationship between the unconfined compressive strength of the water-cement ratio and the cement dosage with the cement-solidified soil [36,37,38,39]. The quasi-water-cement ratio R [40,41] is a parameter that considers the relationship between the water-cement ratio, the mix ratio and the water content, as shown in Equation (1).
(1)$$R=\frac{M}{100}+\frac{100\omega_{n}}{(\omega_{n}+100)a_{w}}$$
In Equation (1), M is the water-cement ratio (%), $\omega_{n}$ is the natural moisture content (%) of the soil and aw is the blending ratio (%).

According to the test results, the unconfined compressive strength q_u_ of DGSC-solidified soil can be linearly related to the quasi-water-cement ratio through 1/R (see Figure 8).
(2)$$q_{u}=K_{E}(\frac{1}{R}-\frac{1}{R_{0}})$$
In the formula, KE is the slope of the straight line and is called the reinforcement coefficient of the DGSC reinforced soil. Numerous studies have shown that the reinforcement coefficient KE of cement-solidified soil increases with its curing period [42,43]. Figure 8 also shows the KE of the DGSC-solidified soil. The intersection of the straight line and the abscissa is 1/R_0_ and R_0_ is the maximum quasi-water-cement ratio. Equation (2) shows that R_0_ of a certain type of soil can be regarded as a constant. Based on the regression analysis shown in Figure 8, it can be concluded that the R_0_ of this type of soil is 4.1.

### 4.2. Relationship between the Consolidation Coefficient (KE) and the Curing Time for DGSC-Solidified Soil

In practical projects, the strength of cement-solidified soil is often tested by 28-day sampling to determine its curing effect. To better understand the development of the solidified soil in the later period, it is necessary to predict its strength. For example, Sakka et al. [42] and Horpibulsk et al. [43] proposed corresponding prediction formulas. In this paper, the prediction effect (2) of 1/R, KE, is used to study the curing effect of the DGSC curing agent and its feasibility in engineering applications. The relationship between KE/K28 and T is shown in Figure 9. Through regression analysis, the results are as follows:
(3)$$\frac{K_{E}}{K_{28}}=0.21+0.3\ln(T-6.25)$$


From Equations (2) and (3), we can obtain the following:
(4)$$q_{u}=[0.21+0.3\ln(T-6.25)]K_{28}(\frac{1}{R}-\frac{1}{4.1})$$


When a certain quasi-water-cement ratio R and its strength coefficient K28 are known, the strength of the DGSC-solidified soil at any curing time can be calculated by Equation (4). Figure 10 compares the measured intensity with the predicted strength and the correlation coefficient is 0.98, which shows a good correlation. Therefore, it is feasible to use Equation (4) to predict the strength of DGSC-solidified soil.

## 5. Conclusions

As the DGSC content increases from 8% to 20%, the unconfined compressive strength of the solidified soil increases from 481 kPa to 3035 kPa. It was found that the higher the DGSC content, the better the curing effect on soft soils. When the water-binder ratio of solidified soil is 0.5, after the 28-day curing period, the unconfined compressive strength of the 15% DGSC content is 1441 kPa and for a15% cement content it is 1335 kPa, indicating that the content of DGSC is higher than that at 5% cement content; so, the DGSC solidification effect is comparable to that of cement. Therefore, using DGSC as a soft soil solidifying agent instead of cement can meet the strength requirements of solidified soil. The early strength of DGSC-solidified soil is low and the potential for an increase in strength in the later stage is high; thus, the early strength of this cement-based material needs to be further studied so that the late strength can be stimulated earlier. When the water-binder ratios are 0.3, 0.4 and 0.5, the 28-day unconfined compressive strengths for a 20% DGSC content are 3035 kPa, 2620 kPa and 2451 kPa, respectively. Additionally, the increase in the water-cement ratio is inversely proportional to the increase in strength. Finally, a regression analysis comparing the measured intensity with the predicted strength shows a correlation coefficient of 0.98; therefore, the strength of solidified soil can be accurately predicted in the later period.

## Figures and Tables

**Figure 1 materials-12-00826-f001:**
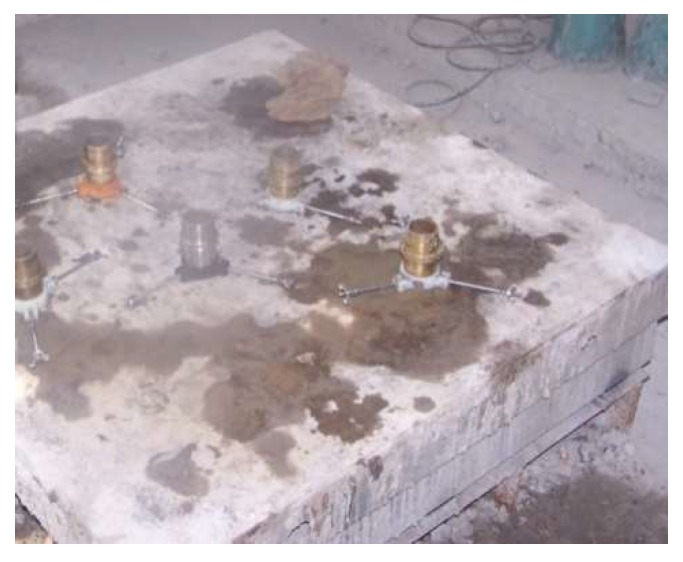
Layered vibration tamping with the shaking table.

**Figure 2 materials-12-00826-f002:**
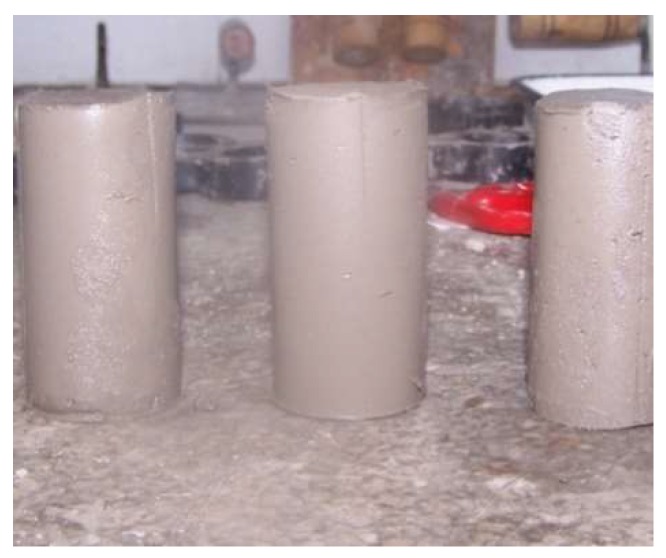
Sample after demolding.

**Figure 3 materials-12-00826-f003:**
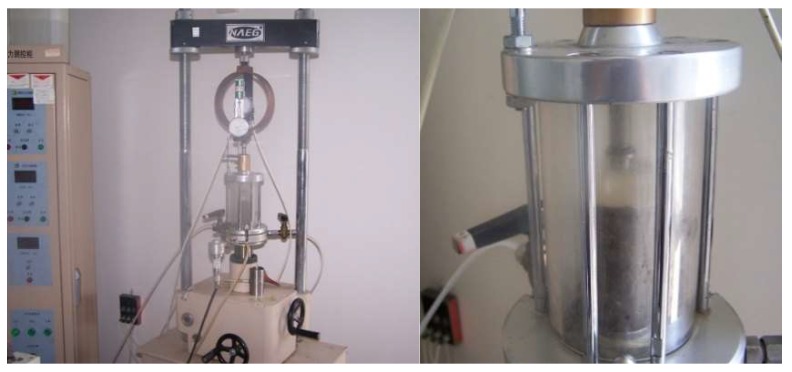
Unconfined compressive strength test by triaxial shear tester.

**Figure 4 materials-12-00826-f004:**
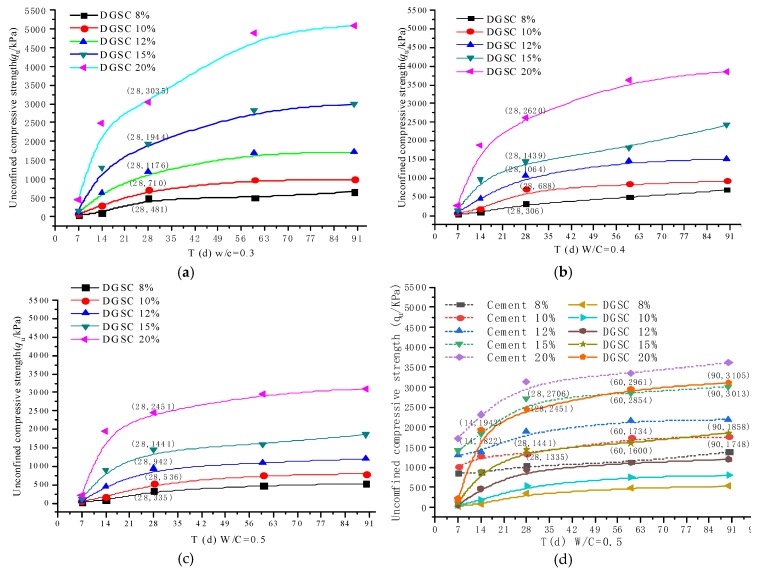
T-qu curves of the solidified clay under (**a**) 0.3, (**b**) 0.4, and (**c**) 0.5 of the water-binder ratio and (**d**) the comparison of cement and DGSC under 0.5 water-binder ratio.

**Figure 5 materials-12-00826-f005:**
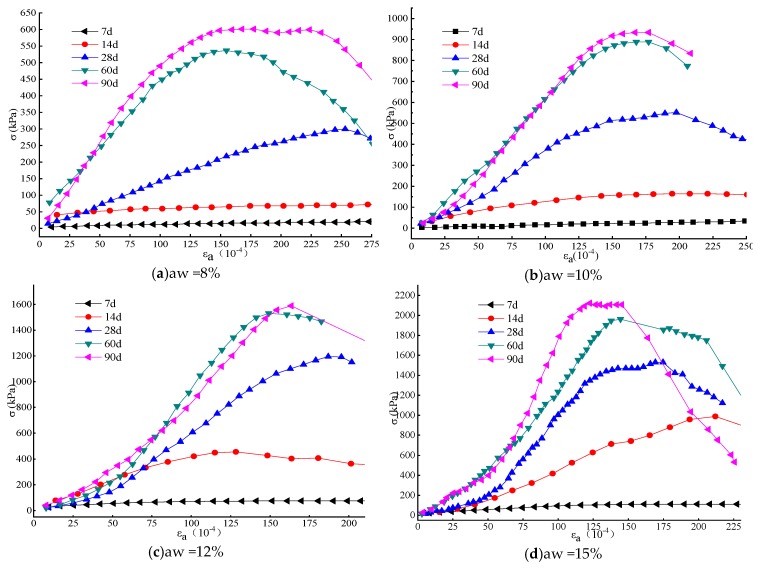
Stress-strain relation curves of DGSC-solidified soil under (**a**) 8%, (**b**) 10%, (**c**) 12%, (**d**) 15%, and (**e**) 20% of the content.

**Figure 6 materials-12-00826-f006:**
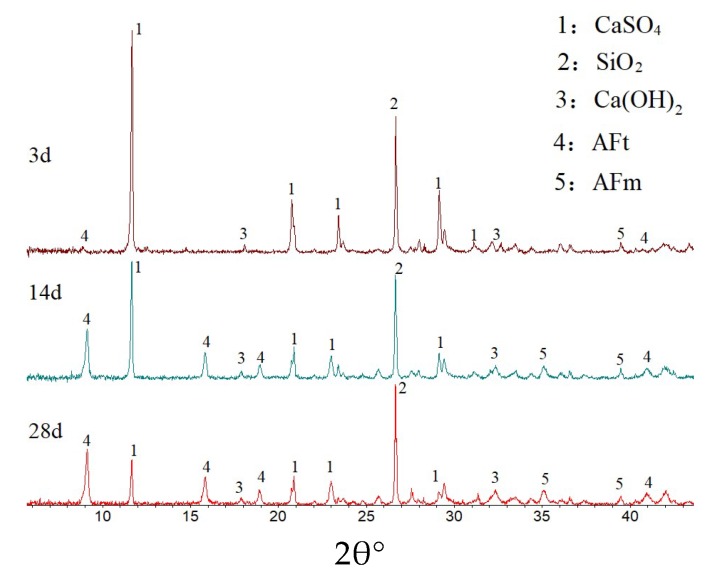
X-ray diffraction (XRD) diagrams of curing agents at different curing times.

**Figure 7 materials-12-00826-f007:**
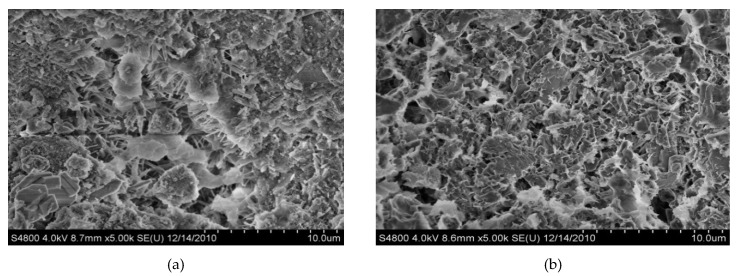
Scanning electron microscopy (SEM) of DGSC. (**a**) at 3 days (5k); (**b**) at 28 days (5k).

**Figure 8 materials-12-00826-f008:**
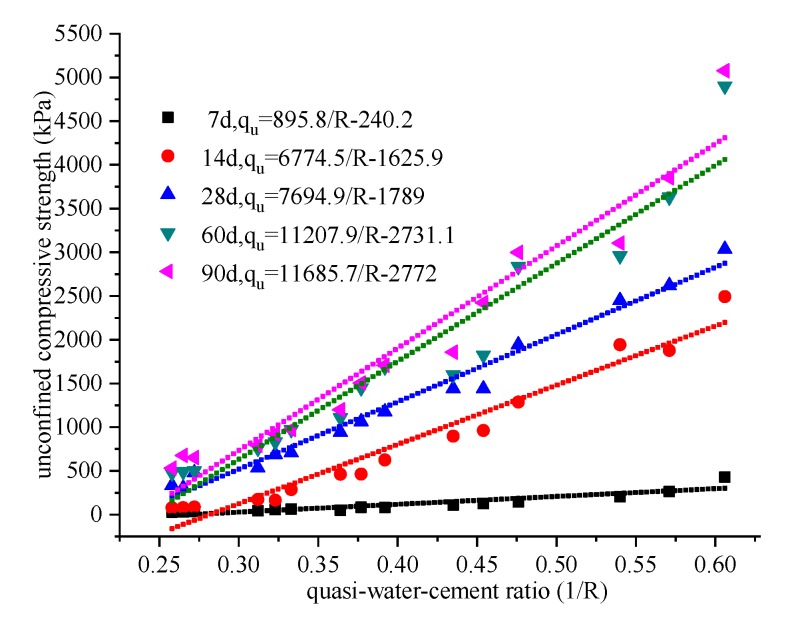
Relationship between the unconfined compressive strengths and the quasi-water-DGSC ratio.

**Figure 9 materials-12-00826-f009:**
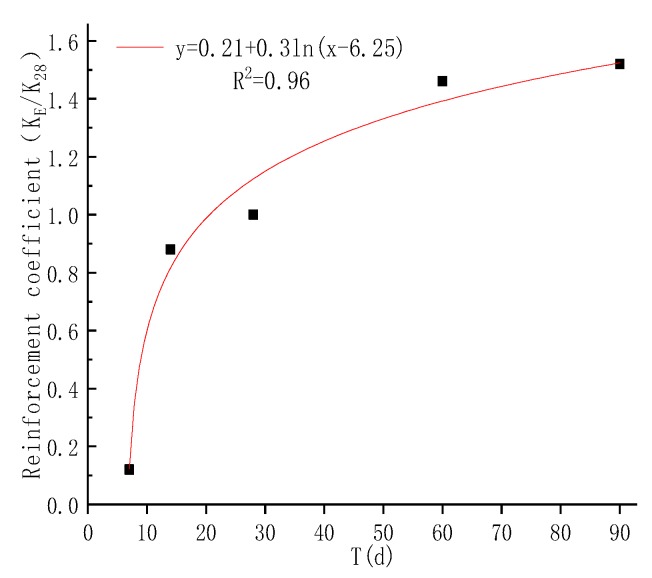
Relationship between KE/K28 and curing time.

**Figure 10 materials-12-00826-f010:**
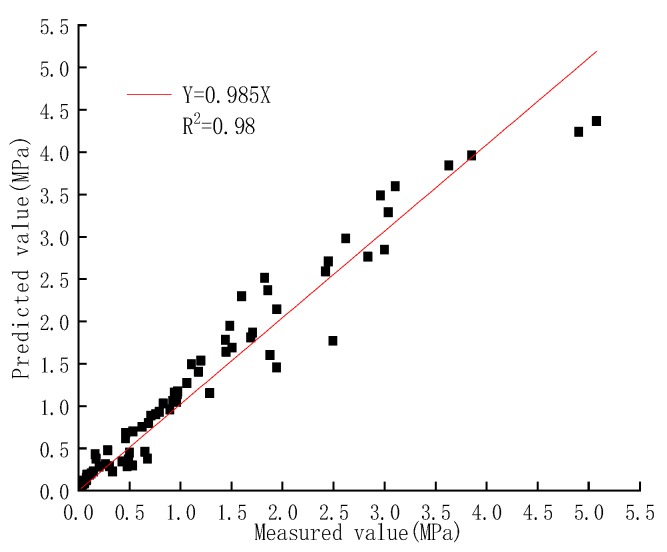
Comparison between the tested and predicted strengths.

**Table 1 materials-12-00826-t001:** Main chemical compositions of the waste residues.

Waste Residue	SiO_2_	Fe_2_O_3_	Al_2_O_3_	CaO	MgO	MnO	P_2_O_5_	SO_3_	IL
SS	9.04	27.23	1.88	41.50	10.24	3.14	1.55	-	1.65
S	32.30	0.20	14.30	39.00	7.70	-	-	1.00	1.89
DG	1.50	0.29	0.80	41.40	0.12	-	-	55.43	-

**Table 2 materials-12-00826-t002:** Physical and mechanical properties of the cement and DGSC.

Curing Agent	T/min		Average Compressive Strength (fc/MPa)		Average Flexural Strength (ff/MPa)	
	Initial Set	Final Set	3d	28d	3d	28d
Cement	135	213	24.70	48.90	4.70	7.30
DGSC	90	295	20.40	44.20	5.50	8.50

**Table 3 materials-12-00826-t003:** Physical parameters of the soil sample.

Soil Samples	Water Content/%	Pellet Specific Gravity (g/cm^3^)	Porosity Ratio	Liquid Limit (ωL/%)	Plastic Limit (ωP/%)	Plastic Index IP	Liquid Limit Index IL
Muddy clay	37	2.73	1.05	37.50	21.10	16.40	0.82

**Table 4 materials-12-00826-t004:** Orthogonal test.

NO	DG/%	A/% Cement Clinker: Sulphoaluminate Mineral	S/%	SS/%
1	1(12)	1(2.50:2.50)	1(30)	53
2	1(12)	2(2.78:2.22)	2(35)	48
3	1(12)	3(2.22:2.78)	3(40)	43
4	2(15)	1(2.50:2.50)	2(35)	45
5	2(15)	2(2.78:2.22)	3(40)	40
6	2(15)	3(2.22:2.78)	1(30)	50
7	3(18)	1(2.50:2.50)	3(40)	37
8	3(18)	2(2.78:2.22)	1(30)	47
9	3(18)	3(2.22:2.78)	2(35)	42

**Table 5 materials-12-00826-t005:** Physico-mechanical parameters of the curing agent with different mix proportions.

NO	Stability	Standard Consistency (g)	T/min		Average Flexural Strength (ff/MPa)		Average Compressive Strength (fc/MPa)	
			Initial Set	Final Set	3d	28d	3d	28d
1	qualified	148	110	380	3.07	6.85	11.69	25.50
2	qualified	147	98	310	3.67	8.00	14.92	43.14
3	qualified	146	90	270	3.97	7.75	14.92	38.39
4	qualified	147	125	404	4.20	8.10	14.75	37.44
5	qualified	144	85	290	5.07	8.40	17.02	44.54
6	qualified	148	130	355	1.87	5.63	7.23	27.15
7	qualified	142	93	315	5.13	8.43	16.12	41.72
8	qualified	147	125	410	3.90	7.90	12.56	41.25
9	qualified	146	135	415	4.00	7.72	12.63	35.39

**Table 6 materials-12-00826-t006:** Content of each admixture of the curing agent.

Category	Incorporation Ratio				
	8%	10%	12%	15%	20%
Dry soil (g)	700	700	700	700	700
Cured material (g)	77	96	115	144	192
Water content (%)	37	37	37	37	37
Water (g) w/c = 0.3	282	288	294	302	317
Water (g) w/c = 0.4	290	297	305	317	336
Water (g) w/c = 0.5	297	307	317	331	355

**Table 7 materials-12-00826-t007:** The effect of the water-cement ratio on the strength of solidified soil.

Curing Agent	w/c	aw/%	qu7/kPa	qu14/kPa	qu28/kPa	qu60/kPa	qu90/kPa
DGSC	0.3	15	146	1287	1944	2836	2999
	0.4	15	127	962	1439	1825	2423
	0.5	15	111	897	1441	1600	1858
Cement	0.5	10	1006	1276	1335	1734	1748
		15	1415	1822	2706	2854	3013

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
