# Peer review of "Study on the Stabilization of a New Type of Waste Solidifying Agent for Soft Soil"

_materials, 2019, doi:10.3390/ma12050826_

Round 1

Reviewer 1 Report

The readers can not understand what was happen and how the components of DGSC were changed after mixing. Please state something in the text about role of components of DGSC for the strength development.

Author Response

Dear  Reviewers:

Thank you for your comments concerning my manuscript entitled “Study on the stabilization of a new type of waste solidifying agent for soft soil” (ID: materials-419501). Those comments are all valuable and very helpful for revising and improving my paper, as well as the important guiding significance to my researches. I have studied comments carefully and have made correction which I hope meet with approval.

 see the appendix for details.

Reviewer 2 Report

Introduction Part is insuficient and needs serious reconstruction. Authors refered on only 12 references which 8 of them are just placed in the bracket without any explanation. Some references doesn't belong in this area of research of stabilized soil.

Previous investigations mentioned in ref 9-12 are not comprehensive and it is not clear on which material they refer to.

In table 3. measuring unit for specific gravity is missing (g/cm3)

In part 2.2. basic information about cement-soil is missing

In the parts 2.2.1. and 2.3.1. for better understanding addition of few photos are necessary.

For better comparison of the results, in the Fig. 1 all values on Y axes need to be the same value (5500 kPa)

In the Part 4. authors refered on 8 references (13-20) but in the reference list those are missing.

Description of axes in Fig.7 needs to be in english. 

Conclusion Part should be explained with more details.

Author Response

Dear  Reviewers:

Thank you for your comments concerning my manuscript entitled “Study on the stabilization of a new type of waste solidifying agent for soft soil” (ID: materials-419501). Those comments are all valuable and very helpful for revising and improving my paper, as well as the important guiding significance to my researches. I have studied comments carefully and have made correction which I hope meet with approval.

 see the appendix for details

Reviewer 3 Report

The paper describes the investigation of mechanical properties with DGSC (desulfurization gypsum, steel slag/furnace slag composite cementitious material) by comparing the unconfined compressive strength of cement treated and with DGSC treated soil. The title of the paper corresponds to the content of the work.
The research presented in this paper represents an interesting supplement to the results of numerous investigation of stabilised soils properties, and is in complete correlation with them. The results well support the study conclusions.
Writing does not correspond to a scientific paper. Correction of English is required. The paper uses unusual terms such as, for example:
- when the age reaches 7d - after the 7-day curing period
- at the cement-soil incorporation ratio of 15% - at 15% cement content (it has to be specified by mass or by volume)
- when the age reaches 28d, the unconfined compressive strength - 28-day unconfined compressive strength
- with the extension of the age – instead of age it will be better to use time or period
Abbreviations are used without explanation.
Section 2.1. Raw materials – rewrite properly for a scientific paper. Full sentences, not just listing.
In Section 2.2.1. it is not necessary to explain every step of the sample preparation to the smallest details. It is enough to describe the preparation of the sample briefly.
Figure 7 - It is necessary to correct the names of the axes written in the Chinese alphabet
The paper deals with an innovative and current topic so I recommend it for publishing, but only after significant corrections of the way of writing and language corrections.

Author Response

Dear Reviewers:

Thank you for your comments concerning my manuscript entitled “Study on the stabilization of a new type of waste solidifying agent for soft soil” (ID: materials-419501). Those comments are all valuable and very helpful for revising and improving my paper, as well as the important guiding significance to my researches. I have studied comments carefully and have made correction which I hope meet with approval.

See the appendix for detail

Reviewer 4 Report

The subject of the article concerns very practical aspects of the use of waste materials for soil stabilization based on cement (desulfurization gypsum and steel slag-furnace slag composite cementitious material -DGSC. The results of the research may be interest to a very narrow public. The introduction has been very modestly developed and requires a significant extension of information on other waste materials used for soil stabilization. The autors do not explain why give up part of the testing of samples contain 20% of the DGSC. The results are graphically very poorly developed. Different scales were used in the comparison charts. There are also Chinese markings on the charts. The article reguires substantive, editorial and linguistic correction. 

Author Response

Dear Editors and Reviewers:

Thank you for your comments concerning my manuscript entitled “Study on the stabilization of a new type of waste solidifying agent for soft soil” (ID: materials-419501). Those comments are all valuable and very helpful for revising and improving my paper, as well as the important guiding significance to my researches. I have studied comments carefully and have made correction which I hope meet with approval.

        See the appendix for detail

Round 2

Reviewer 4 Report

A slight correction of the data presentation should be mode.

In table 5 the setting time should be given in minute.

The paper should be corrected by a native speeker.

Author Response

Dear Reviewers:

Thank you for your comments concerning my manuscript entitled “Study on the stabilization of a new type of waste solidifying agent for soft soil” (ID: materials-419501). These comments are valuable and very helpful for revising and improving my paper, as well as clarifying the significance of my research. I have studied the comments carefully and have made corrections that I hope meet your approval.

see the appendix for details.
